# The Role of tRNA-Centered Translational Regulatory Mechanisms in Cancer

**DOI:** 10.3390/cancers16010077

**Published:** 2023-12-22

**Authors:** Yuanjian Shi, Yipeng Feng, Qinglin Wang, Gaochao Dong, Wenjie Xia, Feng Jiang

**Affiliations:** 1Department of Thoracic Surgery, Nanjing Medical University Affiliated Cancer Hospital & Jiangsu Cancer Hospital & Jiangsu Institute of Cancer Research, Nanjing 211166, China; syj2023@stu.njmu.edu.cn (Y.S.); njmufyp@stu.njmu.edu.cn (Y.F.); qinglin_wangnjmu@outlook.com (Q.W.); gaochao_dong@njmu.edu.cn (G.D.); 2Jiangsu Key Laboratory of Molecular and Translational Cancer Research, Cancer Institute of Jiangsu Province, Nanjing 210009, China; 3The Fourth Clinical College, Nanjing Medical University, Nanjing 210029, China

**Keywords:** cancer, translational regulation, tRNAs, tRNA modifications, tRNA-derived fragments

## Abstract

**Simple Summary:**

The protein translation machinery, of which tRNA is an essential part, is critical in controlling the growth of tumor cells. The development of cancer is also influenced by translational dysregulation. Understanding the rewiring of gene expression at the translational level, which underpins the development of transformational phenotypes during cancer, has made tremendous strides in recent years. We have found that the mode of translation regulation centered on tRNAs affects the translation system in various cancer types in a variety of different ways. In addition to the expression level of tRNA itself, codon and amino acid usage bias, tRNA modification regulation, and tRNA-derived fragments all play important roles. This review discusses recent developments and new insights to elucidate the role of tRNA-centered translational regulatory models in some aspects of carcinogenic and cancer cell activity.

**Abstract:**

Cancer is a leading cause of morbidity and mortality worldwide. While numerous factors have been identified as contributing to the development of malignancy, our understanding of the mechanisms involved remains limited. Early cancer detection and the development of effective treatments are therefore critical areas of research. One class of molecules that play a crucial role in the transmission of genetic information are transfer RNAs (tRNAs), which are the most abundant RNA molecules in the human transcriptome. Dysregulated synthesis of tRNAs directly results in translation disorders and diseases, including cancer. Moreover, various types of tRNA modifications and the enzymes responsible for these modifications have been implicated in tumor biology. Furthermore, alterations in tRNA modification can impact tRNA stability, and impaired stability can prompt the cleavage of tRNAs into smaller fragments known as tRNA fragments (tRFs). Initially believed to be random byproducts lacking any physiological function, tRFs have now been redefined as non-coding RNA molecules with distinct roles in regulating RNA stability, translation, target gene expression, and other biological processes. In this review, we present recent findings on translational regulatory models centered around tRNAs in tumors, providing a deeper understanding of tumorigenesis and suggesting new directions for cancer treatment.

## 1. Introduction

Almost all tumors are characterized by abnormal cell growth and proliferation, which is primarily attributed to changes in gene expression within cancer cells [1]. Recent advancements in RNA sequencing, mass spectrometry, and informatics have provided valuable insights into the molecular mechanisms underlying translational regulation of gene expression, allowing researchers to gain a better understanding of alterations in cancer gene expression. Among these regulatory mechanisms, transfer RNA (tRNA) has gained considerable attention due to its critical role in oncogenesis and translational regulation [2,3]. Protein synthesis, a vital step in gene expression, plays a crucial role in various fundamental cellular processes and is essential in regulating tumor initiation and progression, with tRNA serving as a key player [2]. TRNA, which is mediated by RNA polymerase III and synthesized in the nucleus, plays a pivotal role in ensuring precise protein synthesis by regulating the activation of the protein translation system. Meanwhile, dysregulation of tRNA expression is often observed in cancer cells, leading to translational dysregulation that significantly contributes to cancer development and progression [4]. The substantial impact of tRNA on tumorigenesis and development is well-documented through extensive investigations into its regulation in cancer [5,6,7]. Oncogenes and tumor suppressor signaling pathways, such as inosine phosphate 3 kinase (PI3K)/TORC1, RAS/extracellular signal-regulated kinase (ERK), Myc, p53, and retinoblastoma gene (Rb), participate in the translational regulation of tumors by modulating the synthesis of pol III and tRNA [1,8,9,10]. Additionally, the development of several disorders, including cancer, is greatly influenced by the complex nature of tRNA synthesis, modification patterns, regulation, and function. Numerous creative suggestions for cancer prevention and therapy have emerged from the most recent studies on tRNA in malignancies [11,12]. Currently, all human tRNA isoforms have more than 100 alterations discovered. Small tRNA fragments (tRFs) are produced when tRNA undergoes specific changes in response to particular stressors. TRNAs have a variety of functions in carcinogenesis due to their wide range of alterations and the quantity of generated tRFs [13]. Together, our results highlight the importance of tRNA expression dysregulation and translation error rates in cellular heterogeneity, proteomic variety, genomic instability, and drug resistance in cancer. This review focuses on the current understanding of translational regulation of tRNA in tumors and its implications for cancer research.

## 2. tRNA Influences Translational Regulation of Tumors at Multiple Levels

Correct translational regulation is crucial for cells to carry out their biological functions as it is involved in nearly all cellular processes. The unique proliferation pattern of tumor cells is closely associated with the deregulated state of translation in cancer. Simultaneously, this deregulated state controls the growth of cancer cells and maintains the adaptive capabilities of tumor cells [14,15]. Coordination among various systems that regulate gene expression at the translational level exists at multiple levels, including tRNA copy number, amino acid expression, codon usage bias, and post-transcriptional tRNA modifications. Given that tRNA levels are commonly heightened in different types of cancer, they have been traditionally perceived as housekeeping molecules with little contribution to the constraining stages of translation and gene expression. Nevertheless, emerging evidence suggests that this perspective may be overly simplistic. Both oncogenes and tumor suppressor pathways have the capacity to modulate tRNA synthesis. Furthermore, alterations in the tRNA pool can exert a significant influence on mRNA translation and cellular growth [16]. The human genome contains 61 different sense codons, several of which are synonymous and encode the same amino acids. These 61 codons are translated by 49 distinct tRNA “isoacceptor species”, which carry the same amino acids but have different anticodon sequences. Studies have found that tRNA and isoacceptor species are widely expressed in various human cancers. Alterations in the abundance of tRNA molecules within tumors have the potential to modulate the efficiency of protein translation in cells. Moreover, cancer cells can undergo evolutionary changes to finely regulate the expression levels of multiple promoters involved in cancer progression by modulating tRNA levels. Several studies have demonstrated an association between the overexpression of specific tRNA homologous receptors and increased levels of oncoproteins in genes that are enriched with tRNAs cognate codons [17,18,19,20]. However, the significance of overall tRNA content in translation efficiency and the potential role of codon usage in the coordinated regulation of protein expression remain subjects of debate [21,22,23]. This raises the question of whether there is a connection between alterations in tRNA and the expression of the cancer genome, and whether it is implicated in the development of cancer cell functions [24]. Transfer RNA recognizes specific codons during translation and facilitates the transfer of amino acids to initiate and elongate the polypeptide chain. In humans, the expression of tissue-specific tRNA mirrors the codon usage of tissue-specific proteins, indicating a relationship between the active pool of tRNA and the demand for translation [25,26].

### 2.1. The Effect of tRNA Level (Defined as tRNA^amino acid^_anticodon_) on Translational Regulation in Tumor

A sufficient number of tRNAs is required to maintain the rapid rate of protein synthesis, facilitating the rapid proliferation of tumor cells. Transcriptional control of tRNA genes is reliant on Pol III, and the occupancy of Pol III indicates the level of active tRNA expression. Activation of oncogenic signaling pathways, including Akt, the mammalian target of rapamycin (mTOR), Ras-mitogen activated protein kinase (MAPK), and Myc, along with the loss of the tumor suppressor gene TP53, can promote the transcriptional activation of RNA polymerase III, resulting in an overall increase in tRNA expression [20,24,27,28]. To understand the role of tRNA in cancer cell function, it is important to consider that the carcinogenic lesions that drive the growth and proliferation of cancer cells can induce the expression of tRNA. A recent study has presented a novel extension of this concept, which demonstrates that the usage of codons related to cellular functions is inherently ingrained in the transcriptional programs that govern cellular proliferation in both normal cells and cancer cells. Specifically, the study found distinct and selective expression of tRNA pools between states of cellular proliferation and differentiation, which correlated with the bias in codon usage of genes responsible for autonomous cell proliferation and differentiation support. To illustrate this phenomenon, we have depicted a pattern diagram [18,20] (Figure 1). In addition to global changes in tRNA expression in cancer, studies have revealed that the upregulation of specific tRNAs plays a crucial role in the functioning of cancer cells. For example, overexpression of the initiator tRNA for methionine (tRNA^iMet^) in human breast epithelial cells has been shown to enhance cell proliferation and metabolism. Simultaneously, the overexpression of tRNA^iMet^ resulted in increased levels of other tRNAs, highlighting the complexity of tRNA regulation and suggesting the presence of a feedback regulatory mechanism within the cell [27,29]. These findings suggest that oncogenes possess the ability to selectively induce the expression of target tRNAs, contributing to tumor aggressiveness in a tissue-specific manner, though the exact mechanisms are not yet fully understood. The selective control of specific tRNAs aims to optimize the expression of genes associated with cancer, rather than housekeeping or cell line-specific genes. This is believed to play a role in the development of cancer features, such as sustained proliferation. These observations indicate that changes in tRNA expression in different types of cancer are unique in the context of overall elevation. We analyzed the data from the tRic database and also confirmed this result (Figure 2A). Similar patterns have also been observed in amino acid and codon usage. These findings demonstrate that increased tRNA expression not only fulfills the increased demand for global protein synthesis in tumor cells, but also influences translational control, guiding cell transformation and tumorigenesis [28,30].

### 2.2. The Effect of Amino Acid Level (Defined as tRNA^amino acid^) on Tumor Translation Regulation

The classification of tRNAs is based on the analysis of the amino acids they accept under changing amino acid levels. However, the expression levels of individual tRNAs for each amino acid in cancer exhibit significant variation, despite overall elevated amino acid expression. This expression follows certain patterns. For instance, tryptophan (Trp), arginine (Arg), and cysteine (Cys) tRNA genes are highly expressed in various cancers, while proline (Pro), serine (Ser), and selenocysteine (Sec) tRNA gene expression levels are low [31]. Moreover, there are variations in expression among different types of cancers. For instance, chromophobe renal cell carcinoma (KICH) shows low Pro levels and high phenylalanine (Phe) levels, whereas kidney renal clear cell carcinoma (KIRC) exhibits a completely different pattern [31]. These findings indicate that tRNA expression in tumors undergoes selective shifts at the amino acid level, potentially serving important functions [31] (Figure 2B). Differential expression of tRNAs carrying specific amino acids has been associated with cancer hallmarks. For instance, the upregulation of tRNAs for Arg has been observed in cancer cells [14]. In addition, overexpression of tRNA^Arg^ has been shown to enhance the invasive potential of cancer cells and promote metastasis. This effect may be attributed to the increased stability and translation of genes with a high content of arginine codons [14]. Moreover, tRNAs for Arg have been found to be upregulated in breast cancer. Interestingly, the oncogene TERT protein is also upregulated in breast cancer samples [32]. Furthermore, it has been observed that the frequency of Arg usage by TERT is significantly higher than the average frequency of genomic Arg usage. This finding provides supporting evidence that the overexpression of tRNAs carrying specific amino acids can lead to the overexpression of genes with a high frequency of amino acid usage, thereby overcoming the bottleneck in tumor development. Oncogenes may regulate tumor cell proliferation and survival through this mechanism.

### 2.3. The Effect of Codon Level (Defined as tRNA^amino acid^(codon)) on Tumor Translational Regulation

This phenomenon has also been observed at the level of codon usage bias (Figure 2C). Understanding the interplay between codons and tRNAs in cancer is crucial before analyzing changes at the codon usage bias. On one hand, the abundance of specific tRNAs enhances the translation of mRNAs enriched with the cognate codons of the tRNAs, including tumor-driving proteins such as cell cycle regulators [33]. On the other hand, codon usage bias significantly contributes to the differential expression between genetic analogues with similar amino acid homology, thereby influencing tRNA expression changes. The role of codons in cancer can be summarized in two ways. Firstly, they have the potential to enhance translation output by facilitating the translation elongation of various mRNA subsets. Secondly, translation elongation and translation efficiency are influenced by the choice of synonymous codons, which specify the insertion of the same amino acid, but differ in their decoding properties [21,34,35]. TRNAs can play a causal role in driving cancer progression by modulating distinct gene expression networks in a codon-dependent manner. For instance, the presence of rare codons within the KRAS message can hinder translation and, consequently, lead to reduced protein levels and, to a lesser extent, mRNA levels, resulting in a decrease in oncogenic activity [4]. Moreover, there is a selective shift in codon usage in cancer, primarily manifested by the fact that specific codons can support the translation of particular gene expression programs. Recent studies have discovered that functionally relevant codons are closely associated with transcription programs involved in cell proliferation. The preference for these codons in tumors compared to normal cells aligns with the selective and differential expression of the tRNA pool between cell proliferation and differentiation. Concurrently, the expression of tRNA isoacceptors mirrors this phenomenon. For example, the overexpression of specific tRNA isoacceptors has been observed to be positively correlated with the codon usage patterns of cancer-related genes involved in the cell cycle, extracellular matrix, and transcriptional control in breast cancer cell lines [21]. Furthermore, it has been established that changes in codon usage in various cancer types are closely linked to patient survival time and can frequently serve as a prognostic biomarker [36,37,38,39]. Similarly, by reducing the abundance of high-frequency codons and enhancing codon instability, the invasive and metastatic potential of cancer cells can be reduced.

### 2.4. Unequal Alterations at the tRNA Copy Number, tRNA^amino acid^, and tRN^Aamino acid^(codon) Levels

tRNA is essential in translation as it connects the codon and the cognate amino acid. We analyzed the data from the tRic database and the results showed that in various cancer types, changes in the expression of tRNA copy number, tRNA^amino acid^, and tRNA^amino acid^(codon) may not always align properly due to selective expression inconsistencies (Figure 2A–C). The inconsistent pattern observed may be attributed to the uneven distribution of tRNA molecules and codons. Each amino acid is characterized by a variable number of detectable codons, ranging from one for certain amino acids like Trp and Met, to five for others like Arg, Leu, and Ser. Similarly, each codon is associated with a variable number of detectable tRNA genes [40]. The synergistic activation of the translation system plays a very important role in cancer.

## 3. Regulation of Protein Translation by tRNA Modification

One prominent attribute of the tRNA molecule is its extensively modified status, which significantly influences both its biogenesis and function. Here, we summarize the common types of tRNA modifications [41,42] (Figure 3). In humans, a single tRNA molecule may undergo the addition of 11 to 13 different modifications at various stages of its maturation process, ultimately impacting translation directly [43]. These modifications can include simple methylation and isomerization events, such as m^5^C, m^1^A, Ψ, m^5^U, m^1^G, m^7^G, and I, as well as more complex multistep chemical modifications [43]. The function of each modification is influenced by its position within the tRNA molecule and its chemical composition. For instance, m^5^C is specifically added to certain sites by three distinct enzymes (NSUN2, TRM4P, and DNMT2), with each enzyme playing a different role in tRNA metabolism. The modifications occurring at the wobble position are particularly diverse and commonly optimize codon usage for gene-specific translation [24].

### 3.1. Regulation of Protein Translation by Correct Anticodon Modification of tRNA

The majority of post-transcriptional modifications concentrate on positions 34 and 37 of the tRNA anticodon loop, which is crucial for ensuring accurate protein synthesis during various translational steps like aminoacylation, decoding, and translocation [44]. The anticodon region plays a crucial role in the functioning of tRNA as it enables accurate recognition and binding of the appropriate codon on mRNA. The specificity of the interaction between the anticodon and codon is vital for maintaining the accuracy of protein synthesis, making the function of the anticodon region indispensable. The anticodon region of tRNA undergoes various modifications to optimize its interaction with mRNA. For example, position 34 of tRNA, which forms wobble base pairs with the third position of the codon in mRNA, serves as a key site for modifications that either restrict or facilitate the decoding properties of the tRNA [4]. Additionally, position 37 of tRNA is extensively modified, which is important for promoting efficient anticodon-codon interactions and preventing frameshifting. These modifications typically prevent the formation of nonfunctional intraloop base pairs within the anticodon region, ensuring the maintenance of an open and structured loop that is necessary for binding to the ribosome [45]. Consequently, any defects in these modifications within this region can lead to aberrant protein synthesis by affecting the decoding process. For example, hypomodified tRNAs missing mcm^5^s^2^U34 are incapable of decoding their associated codons efficiently, resulting in ribosomal stalling and disturbances in protein homeostasis [46]. Q_34_ enhances tyrosine translation in mitochondrial tRNA [47]. Similarly, ac^4^C at the tRNA^Met^ wobble location enhances tRNA decoding of noninitiating AUG codons while lowering tRNA-AUG codon affinity, ultimately leading to slower translation and altering protein production. Furthermore, the ‘distal’ shape of ac^4^C aids in the prevention of AUA codon misreading during translation [48]. I_34_, f^5^C, and m^5^C also extend the decoding capability of tRNA [49]. Notably, m^5^C in tRNA regulates translation efficiency and accuracy by optimizing codon-anticodon pairing [50]. The presence of m^7^G modifications in matching tRNAs may affect ribosomal translocation. The lack of METTL1 enhances ribosome mobility on mRNAs, increasing the frequency of m^7^G-tRNA decoding. Therefore, the expression of related modifying enzymes and m^7^G-tRNA modification have an impact on translation efficiency [51,52,53]. The base composition at position 36 influences the kind of modification at tRNA position 37 [54]. By interacting with the initial base of the codon, these modifications significantly increase the stability of codon-anticodon coupling [55,56]. t^6^A is a highly conserved alteration found at position 37 of tRNA that promotes tRNA binding to the A-site codon and effective translocation, assuring translation accuracy and efficiency [57]. Pseudouridine synthetase (Pus1) experiments in S. cerevisiae have shown that Pus1-dependent pseudouridylation is critical in particular decoding processes in vivo. Pus1 deletion significantly increases tRNA^His^ codon misreading of CGC (Arg) [58]. Collectively, these changes affect codon-anticodon binding and, as a result, the protein translation process.

### 3.2. Several Classical Abnormal tRNA Modifications and Cancer

Recent studies have focused on the dynamic control of tRNA alterations in response to cellular metabolite levels and environmental inputs. Anomalies in tRNA modification at any location have been linked to the development of numerous disorders, including cancer [59]. The increased interest in the variety of tRNA alterations has gotten a lot of attention in cancer research. Considering the role of m^7^G [60] and m^5^C [61] modifications on tRNA stability, this review focuses on the role of these two tRNA modifications in tumorigenesis.

#### 3.2.1. Abnormal 5-Methylcytosine (m^5^C) Modification in Cancer

The production of m^5^C at position 38 in the tRNA anticodon loop requires the DNA methyltransferase superfamily member DNMT2. The dysregulation of this change, which is substantially retained in animals, has been connected to the development and spread of cancer [62]. Reduced tRNA stability and genomic integrity disruption brought on by DNMT2 loss [63] may result in cancer. Another m^5^C writer, NSUN2 (NOL1/NOP2/SUN domain family member 2), exhibits high expression in different cancer types and plays a significant role in cancer progression regulation [64]. It has been discovered that ovarian cancer contains an enrichment of NSUN2-modified tRNA, which is associated with greater tumor heterogeneity [50]. Additionally, head and neck squamous cell carcinoma patients with elevated NSUN2 expression had a worse prognosis [50]. To reduce NELFB transcription and overcome TMZ resistance, NSUN6, a different member of the 5-NOL1/NOP2/SUN domain family, mediates m5C change in glioblastoma [65]. On the other side, prostate cancer exhibits a downregulation of NSUN6 expression, making it a potential prognostic marker [66].

#### 3.2.2. Abnormal 7-Methylguanosine (m^7^G) Modification in Cancer

It has been established that the m^7^G tRNA mutation promotes cancer by preferentially increasing the translation of oncogenic genes in tumors. For instance, it has been found that the m^7^G mutation significantly increases the expression of genes involved in the cell cycle and the epidermal growth factor receptor (EGFR) pathway in intrahepatic cholangiocarcinoma (ICC) [52]. Reduced modification and therefore reduced translation of tumor-related mRNAs result from the depletion of the m^7^G modification enzymes METTL1/WDR4, which affects the potential of tumor cell division, colony formation ability, tumor cell migration and invasion ability and tumorigenic potential [67]. Notably, the expression of METTL1 is adversely correlated with the prognosis of patients with breast cancer. Breast cancer cell proliferation, migration, and invasion are decreased when METTL1 is silenced [68] through upregulating EGFR/EFEMP1. Similar to this, elevated METTL1/WDR4 expression in nasopharyngeal carcinoma (NPC) encourages both in vitro and in vivo NPC cell proliferation and metastasis. The upstream transcription factor ARNT promotes the growth of METTL1 [69], which modifies the m^7^G tRNA, activates the WNT/β-catenin signaling pathway, boosts epithelial-mesenchymal transition (EMT), and increases the NPC cells’ resistance to chemotherapy. METTL1/WDR4 are significantly up-regulated in esophageal squamous cell carcinoma (ESCC), and silencing any of these genes lowers m^7^G-tRNA modification. Oncogenic genes that are abundant in the RPTOR/ULK1/autophagy pathway are translated more frequently when the modification is decreased [70].

## 4. Regulation of tRNA Fragmentation on Translation in Cancer

TRNA-derived fragments (tRFs) are among the most abundant non-coding RNAs (ncRNAs) in cancers [71]. These fragments are formed when various ribonucleases cleave tRNA precursors or mature tRNAs [72]. A growing number of these short RNAs have been identified using high-throughput RNA sequencing [73,74]. Despite previously being dismissed as degradation products or random cleavage by-products, mounting evidence indicates that tRFs are functional and are linked to a variety of human illnesses, including cancer [75,76,77,78]. In recent years, the significance of tRNA disruption in malignancies has grown in importance, and aberrant tRFs synthesis in cancer has attracted a lot of interest. Numerous investigations have demonstrated that tRFs may both negatively and favorably regulate global translation [79,80,81].

### 4.1. Sources and Types of tRFs

Traditional tRNAs are distinguished by their length, which ranges from 75 to 93 nucleotides, and by their high degree of conservation. They are made up of four arms: the D-arm, the anticodon arm, the TψC arm, the acceptor arm, and a variable arm. In the nucleus, precursor tRNAs (pre-tRNAs) are initially transcribed from tRNA genes by RNA polymerase III. This is followed by cleavage of the 5′ leader and 3′ tail sequences by ribonuclease P and ribonuclease Z. The pre-tRNAs then undergo intron splicing by tRNA endonucleases, attachment of a CCA sequence at the 3′ termini, and additional modifications during tRNA maturation. Different ribonucleases cleave tRNAs at distinct places, resulting in the formation of various tRFs [72]. MINTbase classifies mature tRNAs into five kinds of tRFs: 5′-tRF, 3′-tRF, 5′-halves, 3′-halves and i-tRF [82,83] (Figure 4).

### 4.2. Global Inhibition of Translation Regulation by tRFs

Under stressful conditions, ANG (angiogenin) enhances the creation of tRNA halves. It should be noted that ANG-induced 5′-tiRNAs (such as 5′-tiRNAAla and 5′-tiRNACys) obstruct translation initiation. These tiRNAs control translation by replacing the cap-binding complex eIF4G/A/F, promoting the creation of stress particles (SGs), and collaborating with the translation suppressor YB-1 [84]. The study also showed that interactions with 5′-terminal TOG moieties, comprising four to five guanine residues, are necessary for the creation of a G-quadruplet (G4) structure, which is composed of four 5′-tiRNA molecules. This structure is necessary for the particular 5′-tiRNAs’ ability to regulate translation (Figure 5A) [85,86]. G4-tiRNAs can bind directly to the HEAT1 domain of eIF4G, impairing 40S ribosome scanning and eventually shutting off translation [87]. 5′-tRFs, like tRNA halves, can impact translational initiation by erroneous changes. The researchers proposed that PUS7-mediated alterations in U8 might lead to the binding of polyadenylate binding protein 1 (PABPC1) by mini TOG (mTOG), which contains the 5′-tRFs. For interactions between eIF4G and the start of cap-dependent translation, PABPC1 is a crucial protein. Protein translation is constrained by the mTOG-PABPC1 complex, which prevents PABPC1 from recruiting as much eIF4F. On the other hand, PUS7 is required for tRF-mediated translational control in embryonic stem cells, which is disrupted when PUS7 is lost [88]. This results in increased protein synthesis and incorrect early dermal determination. Another investigation discovered that 5′-tRF builds up and the rate of protein translation reduces when the cytosine-5 RNA methyltransferase NSUN2 is not present [2].

### 4.3. tRFs Mediate Translational Activation through Ribosomes

In tumors, tRFs can also regulate translation by affecting ribosome function. For example, Leu-CAG 3′ tsRNA, a particular 3′-tRF generated from the tRNA^LeuCAG^, connects with mRNA molecules encoding the ribosomal proteins RPS28 and RPS15 [77] (Figure 5B). This association improves ribosomal protein translation, resulting in improved tumor cell viability [89]. In vivo experiments in animal models, knockdown of LeuCAG 3 ‘tsRNA resulted in significant death of orthotopic hepatocellular carcinoma (HCC) xenografts derived from HeLa cells as well as liver cancer patients. These findings emphasize the role of 3′-tRFs in cancer development [3]. In a different research, it was shown that the 5′-tRF Gln19, which has a size of 19 nucleotides, preferentially binds to the tertiary polysynthase complex (MSC) and promotes translational elongation in HeLa cells via ribosomes and poly (A) binding proteins [88].

### 4.4. tRFs Mediates Translation Regulation by Regulating mRNA Stability

Several investigations have shown that tRFs regulate mRNA stability and translation, offering fresh insight on their function in disease processes [90]. tRFs are related to microRNAs (miRNAs), a type of small non-coding RNA (sncRNA) that modulates mRNA stability by binding to partially complementary sites in target genes’ 3′ untranslated region (UTR), thereby regulating the binding of miRNA-induced silencing complexes [91]. Certain tRFs, like miRNAs, can modify protein translation and alter mRNA stability by directly targeting mRNAs, cleaving partly complementary regions, and interacting with RNA binding proteins (RBPs) [92]. Huang et al. revealed, for example, that tRF/miR-1280 suppresses colorectal cancer (CRC) cell proliferation and metastasis by directly interacting with the 3′ UTR of its target gene, JAG2, and reducing Notch signaling (Figure 5C) [93].

## 5. tRNA-Mediated Translation Models under Stress

The survival and proliferation of cancer cells heavily rely on their ability to adapt to various cellular pressures encountered during tumor formation, including oxidative stress, genotoxicity, metabolic stress, and protein toxicity. These cellular stresses are interconnected with the activation of carcinogenic signaling pathways and significantly impact tumor development and progression [94]. The relationship between stress and translation is manifested by the fact that cancer cells can establish their own adaptive cytoprotective pathways in response to different stress conditions. This protective ability is achieved by regulating translation initiation and elongation [95]. In particular, activation of the stress response pathway not only leads to an overall decrease in protein synthesis but also induces changes in the translation program that promote tumorigenesis. This pathway, especially in cancer cells, regulates the activity of translation initiation factors, thereby inhibiting protein synthesis in the early stages of translation and preventing the formation of damaged and misfolded proteins under specific stress conditions [15]. tRNA modification also plays a crucial role in cancer-related stress. Although the exact mechanisms through which tRNA modification affects tumor function during stress adaptation are not fully understood, emerging evidence suggests that tRNA modification regulates different translation outcomes in response to specific stress conditions, which are associated with phenotypic transformations [87,96]. One notable example is the methyltransferase alkylated DNA repair protein AlkB homology 8 (ALKBH8), which alters the position of specific tRNAs, including selenocysteine tRNAs, under oxidative stress conditions. This modification supports the translation of reactive oxygen species (ROS)-detoxification enzymes, such as glutathione peroxidase 1 (Gpx1), enabling cancer cells to exploit oxidative stress conditions [97,98,99]. To summarize, translational control is crucial for cancer cells under stress. Given the unique metabolic characteristics of cancer cells, hypoxia and other stress conditions commonly occur during tumor development. The translational model of an adaptive stress response presents a novel therapeutic option. The combination of translation inhibitors and traditional pharmacological inducers of cellular stress can synergistically enhance anti-cancer strategies and achieve more effective outcomes.

## 6. Conclusions and Future Perspectives

The understanding of translation has undergone significant changes as research progresses and methods improve. It is now recognized that translation is not merely an uncontrollable process, but rather a regulated network that supports tumor growth and function in the context of cancer. It has become evident that translational regulation serves as a major downstream output of carcinogenic signaling and a central factor in drug resistance observed in several clinically treated cancers [100,101]. Abnormal changes in tRNA can disrupt protein synthesis and contribute to pathological conditions, including cancer. Transfer RNA (tRNA) plays a crucial role in translation. It influences the efficiency of translating cancer-related genes and provides the molecules necessary for tumor metabolism through accelerated translation regulation. In response to stress conditions, tRNA establishes adaptive protection mechanisms in cancer cells [102]. Additionally, tRNA-specific changes can have a direct influence on how proteins are expressed during translation by altering how well tRNA recognizes codons and transports amino acids. The importance of tRNA modification enzymes in this process is highlighted by the fact that they are also implicated in cancer translation. Investigations into how these changes act in various cancer types are now under progress. While tRFs, which are fragments of tRNA, perform comparable tasks to microRNAs, more recent research has shown that they also have a variety of other activities, notably in connection to stabilizing mRNA. Uncovering the role of tRNA in the molecular network of cancer at multiple post-transcriptional, translational, and post-translational stages would benefit from a better knowledge of the formation and functions of tRFs. The coordinated activation of protein translation and changes in translation levels under stress are highlighted by the overexpression of tRNAs and translation regulatory enzymes in cancer. These findings have made it apparent that targeted cancer cell elimination may be possible by focusing on translation initiation. Translation initiation inhibition techniques have already been used in clinical trials examining the effectiveness of cancer therapies. In the future, the identification and application of more precise and effective therapeutic approaches at the translational level will be made possible by the decoding of the regulatory language of cancer genome translation, offering fresh methods to interfere with cancer cell phenotypic transformation within the context of translational control networks [103].

## Figures and Tables

**Figure 1 cancers-16-00077-f001:**
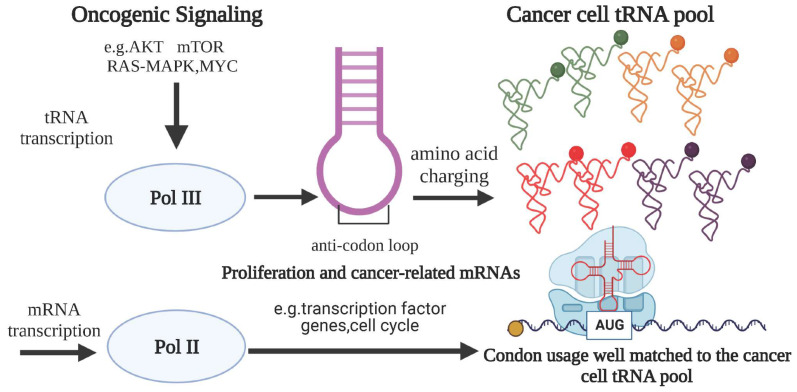
Oncogenic signaling can drive changes in the transcription of tRNA genes. Oncogenic signaling pathways such as AKT, mTOR, RAS-MAPK, and MYC can cause alterations in tRNA gene transcription, resulting in an overall increase in tRNA expression. This, in turn, may encourage the translation of tumor-promoting mRNAs based on codon use bias, in which mRNAs with codons that match the tRNA pool are more likely to be translated. For example, the tRNA pools of cancer cells correspond to the codon use of transcripts involved in cell proliferation and cycle.

**Figure 2 cancers-16-00077-f002:**
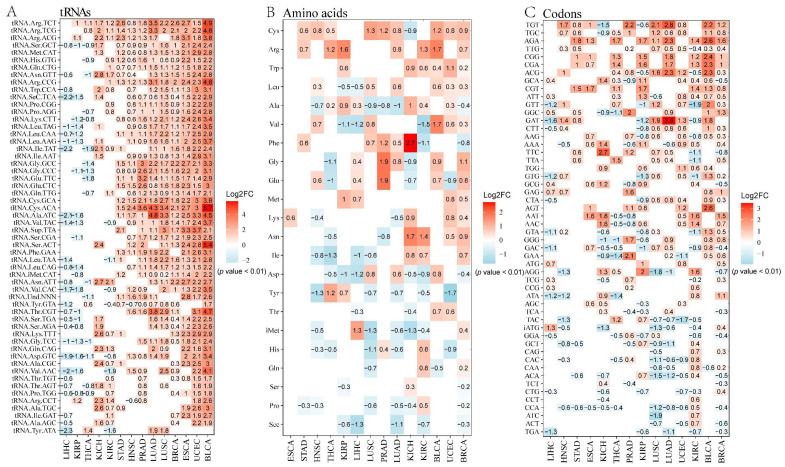
Differential expression of tRNA at three levels in different cancer types. (**A**). Expression characteristics of tRNA gene in different cancer types. (**B**). Expression characteristics of tRNA in different cancer types at amino acid level. (**C**). Expression characteristics of tRNA in different cancer types at codon level. Red represents upregulation, blue represents downregulation, and the specific value shows the log2 fold change of tumor compared with normal. (All dataset were download from tRic database: http://bioinfo.life.hust.edu.cn/tRic/download/, accessed on 10 September 2023). The figure was drawn by R package pheatmap.

**Figure 3 cancers-16-00077-f003:**
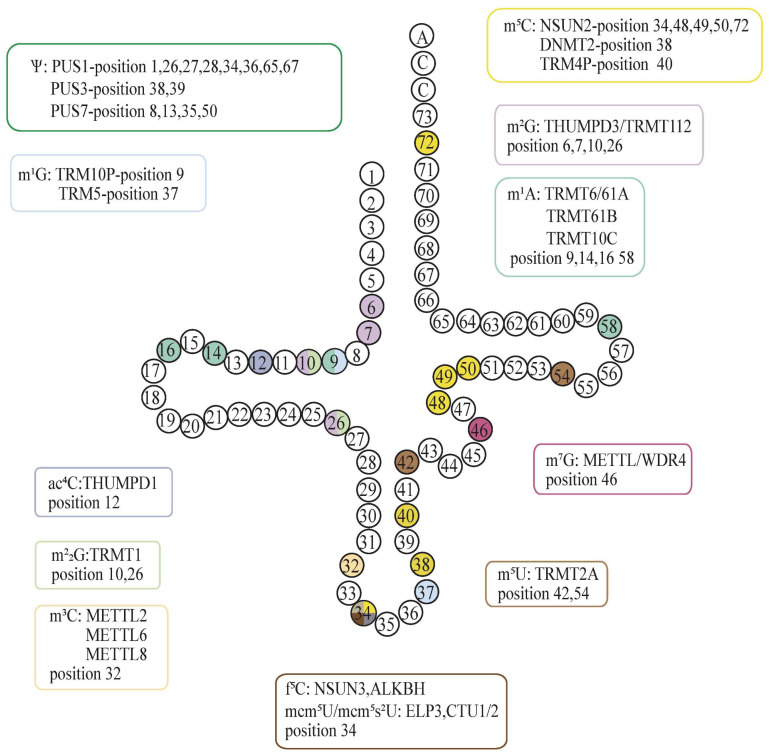
Common tRNA modifications and their modified position discussed in this paper. Ψ: Pseudouridine; m^1^A: N1-methyladenosine; m^2^_2_G: N2, N2-dimethylguanosine; m^5^C: 5-methylcytodine; ac^4^C: N4-acetylcytidine; m^2^G: N2-methylguanosine; mcm^5^U: 5-methoxycarbonylmethyluridine; mcm^5^s^2^U: 5-methoxycarbonylmethyl-2-thiouridine; f^5^C: 5-formylcytidine; m^3^C: 3-methylcytodine; m^7^G: N7-methylguanosine; m^1^G: N1-methylguanosine; m^6^A: N6-methyladenosine; m^5^U: 5-methyluridine.

**Figure 4 cancers-16-00077-f004:**
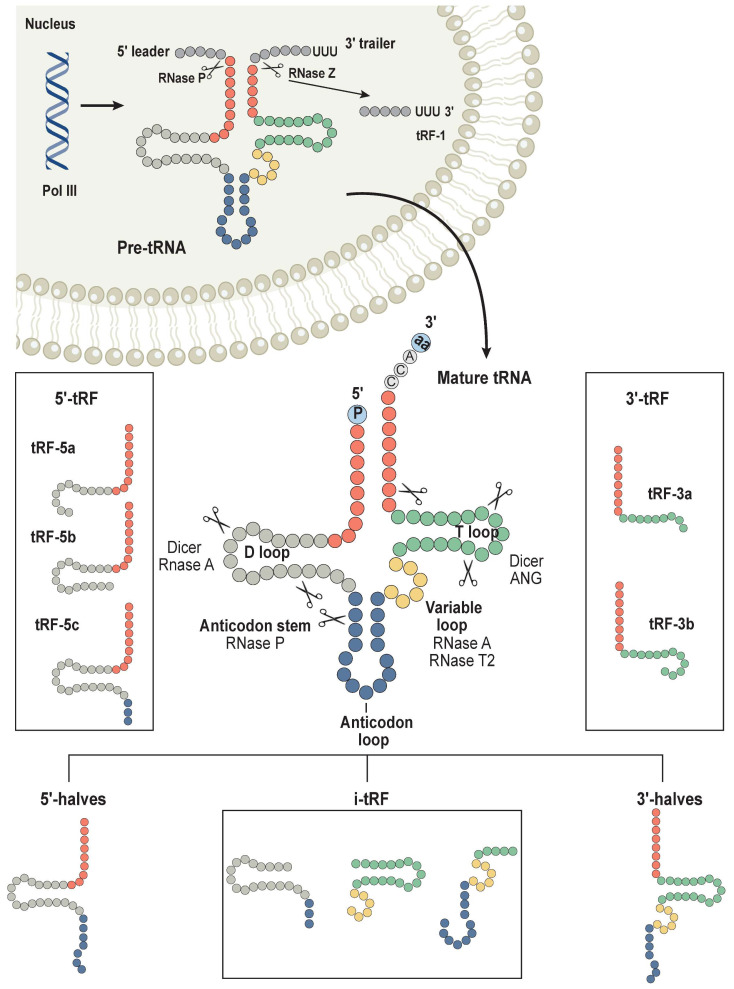
Sources and types of tRFs. Pre-tRNAs are initially transcribed by RNA polymerase III in the nucleus from tRNA genes. Subsequently, RNase P and RNase Z cleave the 5′ and 3′ ends of these pre-tRNAs, respectively. Moreover, tRNA endonucleases are responsible for splicing the introns present in pre-tRNAs, followed by the attachment of the CCA sequence at the 3′ end. These processes contribute to the maturation of tRNAs through additional modifications. Mature tRNAs have different incision sites, which separates them into various types of tRNA-derived fragments (tRFs), including 5′-halves, 3′-halves, 5′-tRFs, 3′-tRFs, and i-tRFs. The cleavage of pre-tRNAs also generates tRFs.

**Figure 5 cancers-16-00077-f005:**
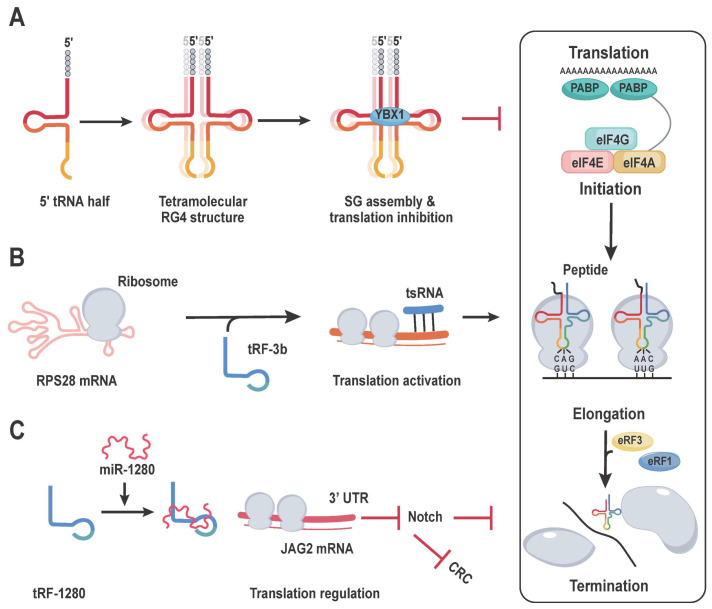
tRF-mediated translational regulation. (**A**) Global inhibition: Angiogenin induces the formation of a 5′ tRNA half with a terminal oligoguanine (TOG) motif. This molecule inhibits translation in a manner that is independent of phospho-eIF2α. The 5′-TOG motif folds into a tetramolecular RNA G-quadruplex (RG4) structure, which induces the assembly of stress granules (SG). This assembly ultimately leads to the inhibition of translation initiation. The involvement of Y-box binding protein 1 (YB-1 or YBX1) is necessary for the assembly of stress granules. (**B**) Translational activation: The LeuCAG 3′ tsRNA binds to RPS28 mRNA, causing a change in its secondary structure. This alteration ultimately leads to the activation of translation. (**C**) Regulate protein translation by affecting mRNA stability: The tRF/miR-1280 molecule effectively inhibited colorectal cancer (CRC) cell growth and metastasis. This inhibition was achieved through direct interaction with the 3′ UTR region of its target gene, JAG2, and subsequent inhibition of the Notch signaling pathway.

## Data Availability

The datasets used or analyzed during the current study are available from the corresponding author on reasonable request.

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
