# Peer review of "The Role of tRNA-Centered Translational Regulatory Mechanisms in Cancer"

_cancers, 2023, doi:10.3390/cancers16010077_

Round 1

Reviewer 1 Report (Previous Reviewer 1)

Comments and Suggestions for Authors

In this resubmission Shi et al, radically rewrote and significantly improved the manuscript previously submitted. They professionally replied to all my comments and criticisms, corrected mistakes, simplified the figures and explained them better. The narrative is also more logical and flows better. I can now recommend this manuscript for publication, subject the following minor, but important suggestions.

1. Figure 3: I think the authors meant to change the depicted tRNA 3’ end from the incorrect UUU to the correct CCA, as they mention in the text and legend, but somehow this did not happen. This should be amended.

2. Lines 205-206: “specific tRNA homologous receptors” – I am not sure what the authors mean here as there are no tRNA, homologous or heterologous, receptors. Do they mean isoacceptors or something similar/else?  Please rephrase.

3. Lines 208-210: Some references to the “various cancer types” in which changes in codon usage “are closely linked to patient survival time and can frequently serve as a prognostic biomarker” are needed here.

4. Lines 337-338: References are needed for the statement: “Numerous investigations have demonstrated that tRFs may both negatively and favorably regulate global translation.”

Comments on the Quality of English Language

There are a few typos here and there, but no major issues.

Author Response

Reviewer 2 Report (Previous Reviewer 3)

Comments and Suggestions for Authors

The Manuscript ID : cancers-2782975 was previously rejected pending revisions. The authors have now revised their manuscript and have responded to the comments. However, most references are still out of date, the author needs to discuss the recent paper as well as the analysis methods of tRNA in this manuscript.

Comments on the Quality of English Language

Minor editing of English language required

Author Response

Reviewer 3 Report (Previous Reviewer 5)

Comments and Suggestions for Authors

The article became significantly better after the correction in accordance with the comments of the reviewers and in my opinion this revised article can be accepted.  

Comments on the Quality of English Language

Minor editing of English language required

Author Response

This manuscript is a resubmission of an earlier submission. The following is a list of the peer review reports and author responses from that submission.

Round 1

Reviewer 1 Report

Comments and Suggestions for Authors

In this manuscript Shi et al. attempt to discuss the role of tRNAs in regulating protein translation in cancer. The paper is badly written, vague and unclear. It fails to build a rational narrative and achieve its objective to explain/clarify/update the role of tRNA in regulating protein translation in cancer. As a result, the reader in the end is non-wiser, and if anything, tired and more perplexed by it.

Importantly, the paper fails to showcase the authors’ knowledge and expertise in the field, if any as I was not able to certify this. Several inaccuracies and mistakes raise ‘red flags’ that have shaken my trust in their proficiency, and considering that I don’t have the time to check every argument they made, I feel that there might be many more mistakes and inaccuracies I did not detect during the time I had to review this work. I therefore cannot recommend this manuscript for publication.

Lastly, I would like to suggest to the authors to check some recent excellent reviews in the field, most of which they did not acknowledge, and evaluate how they can add value to this published work in the future, as there is no point to spend time writing a substandard review with no added value in a topic recently well-covered.

1. Pinzaru, A.M., Tavazoie, S.F. Transfer RNAs as dynamic and critical regulators of cancer progression. Nat Rev Cancer 23, 746–761 (2023). https://doi.org/10.1038/s41568-023-00611-4

2. Orellana, E.A., Siegal, E. & Gregory, R.I. tRNA dysregulation and disease. Nat Rev Genet 23, 651–664 (2022). https://doi.org/10.1038/s41576-022-00501-9

3. Dedon PC, Begley TJ. Dysfunctional tRNA reprogramming and codon-biased translation in cancer. Trends Mol Med. 2022 Nov;28(11):964-978. doi: 10.1016/j.molmed.2022.09.007

4. Suzuki, T. The expanding world of tRNA modifications and their disease relevance. Nat Rev Mol Cell Biol 22, 375–392 (2021). https://doi.org/10.1038/s41580-021-00342-0

Major issues (specific points highlighted as indicative examples):

Figure 2: The data shown here are not from the GtRNAdb as the authors state in the figure legend. I do not know where the authors got them from or even if they have license to use them in this manuscript. This is a major issue.

Figures 1, 3 and 4: These figures seem to contain several typos and inaccuracies, explained below for each one. Also, the authors do not adequately discuss the data presented in these figures in relation to the topic of this review.

Figures 1-5: The lack of references in the figures, as well as in the relevant parts of the text, make it difficult to assess their origin, validity, and relevance in this work.

Line 83: “Previously, tRNA was not thought to be involved in cellular functions.” This is a statement of ignorance and disrespect to the tRNA field, which already had intense activity in the 1950s.

Lines 89-90: “Overexpression of tRNA in tumors can enhance the efficiency of genes involved in cancer development.” This is an example of inaccurate writing, something presented as a fact while it is not, and the absence of relevant references in a topic hotly debated. Noone has argued that tRNA can increase the efficiency of genes, as in gene transcription efficiency. What the authors meant to say is efficiency of protein translation. Does the overexpression of tRNA enhance the translation efficiency of cancer-related and driving proteins? Many would argue not, considering that only few, and not always, tRNAs are overexpressed in different cancers. This is an important topic that this review should discuss, but the authors barely touch upon. Similar inaccuracies and misrepresentations occur throughout the manuscript.  

Lines 94-96: “the functional mechanism of tRNA in cancer is closely linked to changes in codon and amino acid levels”. This does not make sense and is an example of many nonsense sentences in the manuscript. Codons are found in the mRNA and tRNAs do not change them. Amino acids are biochemically synthesised or taken up by nutrition, there is no evidence that the quantity of amino acids is controlled by tRNAs in cancer.

Lines 109-111: “alterations in the expression level of specific tRNAs can affect the overall expression level of tRNAs, thereby regulating the overall translation level of cancer cells”. The authors argue that certain tRNAs can control the expression of other tRNAs (feedback loop in in Pol II transcription?) and this can control the overall protein translation? This is simply wrong and never published. It is also not stated in the references they provided.

Figure 1: Oncogenic signalling (AKT, mTOR, Ras, MAPK, Myc) also affects Pol I and Pol II. This should be shown here. Moreover, the big question is if Pol III (tRNAs), due to oncogenic signaling, is actually contributing more than Pol I (rRNAs) and Pol II (mRNAs) to carcinogenesis than Pol III, and this is not even touched by the authors.

Lines 124-126: “Oncogenic signaling … resulting in the production of particular tRNAs in cancer cells.” Is it true that oncogenic signaling differentially affects specific tRNAs? All the work I have read and that cited in this manuscript show oncogenic signalling affecting the overall production, not individual isoforms. This conclusion seems wrong, or at least unsupported.

Figure 2: The authors do not adequately discuss the data presented in this figure in relation to the topic of this review.

Lines 150-151: “tRNAs for serine, threonine, and tyrosine, which can be easily phosphorylated, are frequently upregulated in cancer” this is inaccurate, unsupported and not cited.

Lines 151-153: “This upregulation may facilitate the charging of these tRNAs with amino acids and contribute to the regulation of multiple cancer signaling pathways.” How? Overexpression of the relevant aminoacyl-synthetase enzymes would be needed for this. Recent papers showed that this is not the case, more often than not. The authors seem to draw unsupported conclusions.  

Lines 157-158: “tRNA overexpression can promote tumor progression by supplying high-demand codons for carcinogenic pathways”. The authors here, and later, seem to confuse codons (found in mRNA) with anti-codons (found in tRNA). What carcinogenic pathways do the authors refer to? Protein translation?

Lines 163-165: “certain “rare” codons that were previously underexpressed in oncogenes are substituted with more frequently used codons, which can promote oncogene expression and tumorigenesis.” Another example of confusing codons with anticodons. Unless they try to discuss specific codon-changing mutations in a convoluted way. This kind of writing is confusing and does not make sense.

Lines 183-185: “Changes in the expression of tRNA, amino acids, and codons may not always align properly due to selective expression inconsistencies. These three components may operate as oncogenes when universally elevated in malignancies; conversely, when consistently down-regulated, they may function as tumor suppressor genes” This is an example of authors not really understanding the definitions of tumour suppressors and oncogenes and the criteria for being one.

Lines 188-185: “As a result, our data suggest that changes in the overall quantities of tRNAs, amino acids, and codons play a key role in triggering particular tumor processes”. Is this copied-pasted from another text? The authors do not present any of their own any data in this work. What particular tumor processes do they mean? It’s unclear.

Lines 194-205: What do the authors feel they need to discuss the wobble effect? This has been resolved in the 1960s.

Lines 205-226: The authors should not just list tRNA modifications. They should explain them and clearly discuss their significance within this review topic.

Lines 245-247: “Reduced tRNA stability and genomic integrity disruption brought on by DNMT2 loss[42] may result in cancer. Another m5C writer, NOL1/NOP2/SUN domain family member 2 (NSUN2), is also carcinogenic in various cancers[43].” To my knowledge, there is no evidence for these being carcinogenic or leading to cancer. To be implicated or found in cancers is very different to actually have the potential to drive carcinogenesis. This is another example of inaccurate and misleading writing.

Lines 277-278: “According to Lee et al., tRNA-derived fragments (tRFs) are the second most abundant small RNAs after miRNAs[51].” It seems like the authors completely ignore the expression of tRNAs, which is the most abundant small RNAs, accounting for about 5-10% of all cellular RNA.

Figure 3: What is 5’ Lesder? Do they mean leader or 5' end? What is 3’ Tailer? Do they mean 3' tail? Why the 3’ has UUU? Should it be CCA? What are the SHOT-RNAs? This is a complicated figure and barely discussed in the text. If the elements of the figure are not discussed, then why are they shown and is this figure at this detail needed? Where did the authors draw the information for this figure(?) - there are no citations. I cannot verify that what is in this figure is correct.

I could keep commenting and providing feedback that will be ignored, but I think I have already made clear my views as to the overall quality and publication suitability of this review.  

Comments on the Quality of English Language

The English level is acceptable, but the authors fail to write high quality text. It often feels that they just list seemingly relevant and often irrelevant details from published papers, without explaining the significance of the information properly. As a result they fail to build an easy to follow, engaging story that makes sense.    

Reviewer 2 Report

Comments and Suggestions for Authors

The study titled "The Role of tRNA-Centered Translational Regulatory Mechanisms in Cancer" takes an in-depth look at the essential role of tRNA in the protein translation machinery. This review highlights recent advances and provides new perspectives on the central role of tRNA-centered translational regulatory mechanisms in carcinogenesis and cancer cell dynamics. This machinery is central to the modulation of tumor cell growth, with dysregulation of translation playing an important role in cancer development. Recently, remarkable progress has been made in understanding how gene expression is reconfigured at the translational level, contributing to the emergence of transformative phenotypes in cancer. Research shows that translation regulation, when centered on tRNAs, affects the translational system in multiple ways in different cancer types. Factors such as the expression level of tRNA, codon and amino acid usage biases, regulation of tRNA modifications, and the presence of tRNA-derived fragments collectively influence this mechanism. This is very interesting. The paper is well written.

Reviewer 3 Report

Comments and Suggestions for Authors

The authors highlight the importance of tRNA expression dysregulation and translation error rates in cellular heterogeneity, proteomic variety, genomic instability, and drug resistance in cancer. This review focuses on the current understanding of translational regulation of tRNA in tumors and its implications for cancer research. Albeit, I consider these findings to provide new insight into cancer-related fields, I still have some suggestions.
1, Most figures are highly professional; however, the authors should guide the readers to the meaning of the images appropriately; otherwise, it will likely cause misunderstandings. Therefore, I suggest the author consider revising these figures and legends again.

2, In Figure 2, the author presented the differential expression of tRNA at three levels in different cancer types. However, it would be much better if the authors could provide some Workflow or Scheme for this analysis, I suggest that they can take a look at the recent paper in MDPI (PMID:  35563422, 36677020, 34834441)

3, There are few typo issues for the authors to pay attention to; please also unify the writing of scientific terms. “Italic, capital”? Please double-check superscripts and subscripts for the whole manuscript.

4, Most references are out of date, the author needs to discuss the recent paper as well as the analysis methods in this manuscript.

5, The font is too small for the current figures; meanwhile, the manuscript also needs English proofreading.

Comments on the Quality of English Language

Editing of English language required

Reviewer 4 Report

Comments and Suggestions for Authors

The manuscript titled "The role of tRNA-centered translational regulatory mechanisms in cancer" presents a comprehensive review of the role of transfer RNAs (tRNAs) and tRNA fragments (tRFs) in the context of cancer biology. The idea behind the manuscript is intriguing and adds value to the existing literature on the molecular underpinnings of cancer.

Specific Comments:

Presentation:

The manuscript is well-crafted, providing a concise yet thorough overview of the subject matter.The language is precise, and the flow of information is logical and easy to follow. This clarity enhances the impact of the research presented.

Content Accuracy:

No significant errors were detected upon initial review, suggesting that the manuscript has been carefully prepared and is based on a solid understanding of the relevant scientific concepts.

Figures and Illustrations:

The figures included in the manuscript, particularly Figure 2, are of high quality. They effectively complement and elucidate the text, which enhances the reader's comprehension of the material. Figure 2 stands out as exceptionally informative and really enjoyed the concept behind it.

Overall, this manuscript appears to be a valuable addition to the scientific literature on cancer research and tRNA biology. 

Reviewer 5 Report

Comments and Suggestions for Authors

Studying tRNA-centered translational regulatory mechanisms in cancer is very important because of its critical role in tumorigenesis. This review examines recent findings on this topic and discusses the current understanding of tRNA translational regulation mechanisms in tumors and its implications for cancer research. The review is written competently and I think this work will be of interest to the readers of the journal. I recommend this manuscript for publication after considering the following points.

In my opinion, the section "Simple Summary" is not necessary. Some data from the "Simple Summary" can be used in the Introduction and in the Abstract sections. The Abstract can be revised taking into account the removal of the Simple Summary. Please add to the introduction more references on this topic including recent books and chapters on cancer. It would be appropriate to add the section "Conclusions" at the end of the work, which outlines the main current findings on this topic and conclusions of the review (for example, "Conclusions and Future Perspectives" instead of the last section "Discussion and future perspectives").

Comments on the Quality of English Language

Minor editing of English language required